# eXnet: An Efficient Approach for Emotion Recognition in the Wild

**DOI:** 10.3390/s20041087

**Published:** 2020-02-17

**Authors:** Muhammad Naveed Riaz, Yao Shen, Muhammad Sohail, Minyi Guo

**Affiliations:** 1Department of Computer Science and Engineering, Shanghai Jiao Tong University, Shanghai 200240, China; nomi_malik@sjtu.edu.cn (M.N.R.); guo-my@cs.sjtu.edu.cn (M.G.); 2Department of Automation, Shanghai Jiao Tong University, Shanghai 200240, China; sohail_sjtu@sjtu.edu.cn

**Keywords:** deep learning, emotion classification, CNN, FER, embedded devices, CK+, RAF-DB

## Abstract

Facial expression recognition has been well studied for its great importance in the areas of human–computer interaction and social sciences. With the evolution of deep learning, there have been significant advances in this area that also surpass human-level accuracy. Although these methods have achieved good accuracy, they are still suffering from two constraints (high computational power and memory), which are incredibly critical for small hardware-constrained devices. To alleviate this issue, we propose a new Convolutional Neural Network (CNN) architecture eXnet (Expression Net) based on parallel feature extraction which surpasses current methods in accuracy and contains a much smaller number of parameters (eXnet: 4.57 million, VGG19: 14.72 million), making it more efficient and lightweight for real-time systems. Several modern data augmentation techniques are applied for generalization of eXnet; these techniques improve the accuracy of the network by overcoming the problem of overfitting while containing the same size. We provide an extensive evaluation of our network against key methods on Facial Expression Recognition 2013 (FER-2013), Extended Cohn-Kanade Dataset (CK+), and Real-world Affective Faces Database (RAF-DB) benchmark datasets. We also perform ablation evaluation to show the importance of different components of our architecture. To evaluate the efficiency of eXnet on embedded systems, we deploy it on Raspberry Pi 4B. All these evaluations show the superiority of eXnet for emotion recognition in the wild in terms of accuracy, the number of parameters, and size on disk.

## 1. Introduction

Emotion classification is an essential feature of human intelligence which enables us to communicate appropriately. For communication between humans and machines, the role of emotion classification is even more influential [1]. Expression classification has a wide range of real-world applications such as autism detection, depression detection, retail shopping (customer satisfaction), educational sectors, and pain assessment [2]. These applications have brought a lot of researcher’s attention towards emotion recognition, and many important works have been presented. All the mentioned applications take frontal face real-world images as input, which are facial expressions captured in a specific environment. Hence, facial expression recognition (FER) in the wild got much attention for further research.

In the past, FER models based on handcrafted features [3,4,5] did not get much attention, since such models usually yield less accuracy as compared to modern deep learning models. In Reference [5], the author proposed the idea of double stage multi-task sparse learning models to differentiate face parts. Alia et al. [4] introduced the Normalized Cross-correlation Coefficient (NCC)-based network for the classification of frontal face expressions. Although, these methods proved effective for frontal faces, they cannot get good performance for images in the wild.

With the recent evolution of deep learning and Convolutional Neural Networks (CNN), various models have shown effective results in the field of computer vision. Most of the crucial advances inspired by AlexNet [6] and VGG [7] have been made [8,9,10,11,12,13,14,15,16,17,18,19], which use end-to-end approaches to classify a given image using facial expression. The typical pattern among these works is the addition of extra layers as well as increments of additional neurons per layer for achieving higher accuracy. This, on the one hand, has provided better accuracy but, on the other, network sizes have grown too much in width (number of layers) and height (number of features per layer). These large networks [7,20] not only suffer from a higher probability of overfitting but also require extra computational power [21].

As mobile and embedded devices are becoming ubiquitous in daily lives, we need shallower networks that can rely on less computing power and memory and denies the trade-off of accuracy vs. efficiency. Recently, Shao et al. [18] built a shallow network (light-CNN) based on depth-wise separable residual convolutions for FER in the wild. Their proposed light-CNN has a small number of parameters, but their network did not achieve higher accuracy. Therefore, this paper proposes a lightweight parallel feature extraction-based model having better performance on emotion classification and is suitable for embedded and live applications.

The next problem that is facing deep CNN models is data, which is the most critical problem in deep learning. Facial expressions can vary from one person to another because of variations in poses, regions, genders, and environment. In the present, the dataset for FER in the wild is limited as compared to frontal face databases. As a solution to this problem, there are many data augmentation techniques [22,23,24,25,26,27,28] developed for increasing the number of samples. In this paper, we also use some of these techniques for increasing the number of training samples, which helped us in achieving better performance for emotion classification in the wild.

Finally, based on the above analysis, we developed a parallel feature extraction-based lightweight CNN for facial emotion recognition in the wild. The proposed network is highly motivated by the structures of References [7,21]. A wide range of experiments is performed on the (publically or upon request) available emotion classification datasets: FER-2013, CK+, and RAF-DB. Evaluation of eXnet in comparison to benchmark networks is also presented, which shows the superiority of the proposed eXnet. In the end, the trained model is installed on the Raspberry Pi 4B embedded system to check the efficiency of proposed models. The output of trained eXnet against face image is shown in Figure 1. The results of extensive experiments show that our models also demonstrate vivid results on real-time devices.

Our contributions for this paper are follows:We propose a novel, lightweight, and efficient CNN-based method “eXnet” for facial expression recognition in the wild.In order to enhance the data and to stabilize the training process, we also utilize modern data augmentation/regularization techniques:eXnet + cutout [22] (eXnet${}^{\mathrm{cutout}}$)eXnet + mixup [23] (eXnet${}^{\mathrm{mixup}}$)eXnet + cutout+ mixup (eXnet${}^{\mathrm{cm}}$)For quantitative analysis, we provide three types of evaluations:Ablation evaluation: variations among the structure, optimization methods, and loss functions have been made to fine-tune the method. All variants of eXnet have been thoroughly explained in Section 5.Benchmark evaluation: comparison of our method with the benchmark methods prove the efficiency of our work as our approach acquires higher accuracy with much less number of parameters.Real-time evaluation: installation of pretrained weights of eXnet on Raspberry Pi 4B bespeaks that eXnet can recognize emotions in the wild more quickly in comparison with existing approaches.

## 2. Related Work

Facial expression classification has been an active field for many years. A detailed and complete description of facial expression and analysis is given by References [29,30]. In Reference [31], the author gave a brief history of the importance of FER. They gave a complete review of the published number of papers in research journals and conferences from 2006–2019. Recently, Huang et al. [32] published a survey paper on emotion classification that gave the state-of-the-act results of more than eight famous facial expression datasets till 2019. According to this survey article, Reference [19] presents a deep learning method based on facial action units, and they achieved 72% accuracy on the FER-2013 dataset.

Glorot et al. [33] described that every facial behavior is directly associated with some emotion. Facial expressions belong to some human functions like mental state; body language; and, most importantly, voice. Recently, researchers have used voice recognition from videos for emotion classification by combining facial features with audio features in temporal and nontemporal modes, reporting better results [34].

Arriaga et al. [35] was inspired by the VGG [7] and GoogleNet [21] models and used a combination of both models for the derivation of the shallower (smaller size) final model. The author trained their model on the FER-2013 dataset and achieved only human-level accuracy of 65% in a lightweight model.

Jain et al. [17] also designed a deep neural network with residual blocks for the classification of six emotions. The author selected the CK+ and Japanese Female Facial Expression (JAFFE) datasets for the training of the network, claiming up-to-date accuracies of 93.24% and 95.23% on both datasets, respectively. For the network design, two residual blocks, which placed after a combination of 2 convolutional layers with a pooling layer, were used, which resulted in top-of-the-line results.

Liu et al. [36] attempted to apply CNN on facial expression classification. They proposed three different types of networks, and each of the networks performed better on some specific classes of emotion as compared to the other two networks. By using a joint scheme on these three models, the author got a final accuracy of about 65% on the FER-2013 dataset.

Tautkute et al. [10] recently claimed the best results on the CK+ and Indian Spontaneous Expression Database ( ISED). The presented approach used deep align network for landmark detection, which further led to the classification of facial expressions. The suggested network was trained on AffecNet (a larger dataset from all previously available datasets for emotion classification). Furthermore, cross- validation was also applied to other publically available datasets, which resulted in improved accuracy.

Recently, Shao et al. [18] presented three types of different models based on CNN. Out of the three, one was a shallow network (Light_CNN), which was purely designed for hardware constraints. However, their shallow network can only achieve 68% on the FER-2013 and 92.86% on the CK+ datasets, which is marginally outperformed by our robust approach eXnet.

Mehta et al. [37] gave a comparative analysis about the intensities of emotions. They use different types of hand-crafted features (Histogram of Oriented Gradients (HOG) and Local Binary Pattern (LBP)). For the classification of emotion, Support Vector Machine (SVM) is used, which gave better intensities about different types of emotions.

Li et al. [38] showed in his survey that the best accuracy is only 70% on the FER-2013 dataset using pure CNN architecture. Sang et al. [9] achieved 71% accuracy on the same dataset, but their model is not purely based on CNN. They used the joint model of CNN and Support Vector Machine for achieving this accuracy. Agarwal et al. [14] gave an idea about how to choose kernel size and the number of filters for a CNN for the problem of emotion classification. In his work, the author proposed two types of models, one having constant and the second having the variable size of the kernels and numbers of filters. By using these lightweight networks, the author achieved human-level accuracy (65%) on the FER-2013 dataset without using fully connected or dropout layers.

In Reference [39], the author used a multi-column CNN method for emotion classification using Electroencephalogram (EEG) signals. By using EEG signals, the proposed model classifies it in the two types of emotions (valance and arousal). Recently, a number of researchers are attracted toward detection of human emotional states (valance and arousal) through different types of physiological signals. Santamaria et al. [40] presented a comparative study between manual and automatic (using CNN) feature extractions from biomedical signals. Experiments of their proposed approach proved that CNN-based features achieved better results as compared to traditional machine learning methods.

It is well known that data augmentation is the backbone for the successful outcomes of deep learning applications expanding from image classification [6] to speech analysis [41]. To improve the generalization of CNN, all the techniques in image classification uses translation, rotation, flipping, resizing, random erasing, and cropping [7,21].

Zhu et al. [42] explained the technique for data augmentation using CycleGan [43] by employing squared loss as an adversarial loss for the generation of additional images for the training process of CNN that heightened 5–10% accuracy of emotion classification.

Recently, 3D-CNN is booming in the field of computer vision. Salama et al. [44] used 3D-CNN for extracting spatiotemporal features in the EEG signals for emotion classification. They conducted experiments on the DEAP (Dataset of Emotion Analysis using the EEG and Physiological and Video Signals). Their proposed method achieved state-of-the-art recognition accuracies of 87.44% and 88.49% for valence and arousal classes, respectively.

All of the above works achieved significant performance over the traditional practices on emotion classification, but all of these models have some limitations either related to model size or accuracy. In this work, we considered these limitations by building CNN-based frameworks achieving regular performance for the same problem. Additionally, we also show the results of our shallow network eXnet on an embedded system for better representation of performance on real-time devices.

## 3. Proposed Model

We propose eXnet, a shallow network (less number of layers and parameters) offering the higher accuracy as compared to existing deep neural networks for emotion classification. This reduction in the number of layers provides the benefit of fast performance on hardware-constrained systems, and less number of parameters allow the network to gain more generalization under Occam’s razor framework [35]. We also use some famous data augmentation strategies without influencing the size of network to enhance the performance of our network. All the details about eXnet and data augmentation techniques are provided next.

### 3.1. eXnet: Expression Network

Broadly, we divide the working of eXnet into three stages based on the type of feature extraction. The basic feature extraction stage takes an input of 48 × 48 × 3 and extracts the primary features which are forwarded to the next stage for intermediate feature extraction. However, this extraction is performed in a parallel manner providing sparsity to our network. The last stage is responsible for final feature extraction and classification of those features into one of the seven possible outcomes.

#### 3.1.1. Basic Feature Extraction

As shown in Figure 2, this stage contains two convolution layers, followed by max pooling and batch normalization [45]. Throughout our model, each convolution layer is followed by batch-normalization and Rectified Linear Unit (ReLU) activation function hence represented as CBR (Convolution Batch-normalization ReLU). With the motivation of the lightweight network, we perform max pooling in each stage. We start our network with the convolutional layer having kernel size of 3×3 as suggested in Reference [7].

#### 3.1.2. Intermediate Feature Extraction

Inspired by the performance of Inception Net [21] for parallel feature extraction, we utilize the same approach for our method. We design two identical ParaFeat blocks, which serially process input, each block containing two different routes for input. Route A contains two convolutions, a 1 × 1 followed by a 3 × 3, and on route B, input first goes through max pooling of 3 × 3 then follows the precisely same convolutions as route A. To reduce the number of parameters on both routes, we utilize 1 × 1 convolution. The effect of 1 × 1 convolution on parameter reduction is also depicted next (Section 5). This reduction in the number of parameters not only makes our network shallower but also allows it to be more general and less prone to overfitting on small datasets, i.e., CK+. At this step, the reduced feature map of route B is concatenated with the feature map of route A for a more distinctive representation of the input. Both blocks end with max pooling of 2 × 2 for downsampling of the image. This part performs the extraction of required features from images as well as downsampling; therefore, this section is the most important part in the whole eXnet architecture.

#### 3.1.3. Final Feature Extraction

In this stage, we reduce the number of parameters drastically to make our network even more sparse; therefore, the output from the intermediate feature extraction stage goes through 1 × 1 convolution followed by max pooling. The max pooling is repeated again but with the convolution of 3 × 3. Finally, we use a combination of a Global Average Pooling (GAP) [46] with two fully connected layers in sequence for classification of extracted final features. The purpose of using GAP is to limit the number of parameters, and fully connected layers are used for the translation of extracted features into one of the seven basic emotions. The last full layer acts as softmax layer that provides us with the probabilities for each emotion, and the maximum likelihood among all is classified as given by the following equation:(1)$$Classified_{emotion}=max\left(P\left(\frac{Y_{i}}{x}\right)\right)$$
where i∈[0,6] is emotions classes and *x* is given input image.

For the classification of true emotion, here, we use a famous classification loss function named cross-entropy loss, which is given by the following equation:(2)$$L_{CrossEntropy}=-y.\log y^{{}^{\prime }}$$
where *y* is a vector of true labels and $y^{{}^{\prime }}$ represents the estimated emotions.

### 3.2. eXnet${}^{\mathrm{cutout}}$

Deep learning networks should have the ability to learn dominant representations that are important in handling complicated learning spaces. To capture such depictions, required network capacity leads the network towards overfitting. Training needs a strong regularization of a network for generalization to handle these types of issues. Reference [22] proposed a simple regularization technique by cutting out a random portion of fixed size without manipulation of feature maps of visual input during the training of the network. Unlike dropout and its variants, any rescaling of weights at test time is not performed. Using the cutout technique for the training of our eXnet not only enhances the performance of our network but also allows the eXnet to focus on complementary and less prominent features. While testing, cutout encourages the proposed model to mask more minor features rather than to rely only upon a few significant features.

### 3.3. eXnet${}^{\mathrm{mixup}}$

Deep CNN has proved influential in vision tasks, but sometimes, such networks show unwanted outcomes (i.e., memorization and sensitivity) when the image is slightly changed from training samples. Reference [23] introduced a useful technique that trains a CNN on convex combinations of pairs of samples and their respective labels, which allows CNN to favor simple linear behavior among training samples. The term mixup suggests training a model on a mixture of pictures from a training set. For emotion classification, we choose two images from the mentioned datasets. Instead of feeding eXnet with the raw image, we do a linear combination of these images and then forward it to our network. In simple, two pictures and their corresponding labels are randomly selected from the training samples for simple random weighted summation. Mathematically, the mixing up of images (*I*) and its respective labels (*L*) is given in Equations (Equation 3) and (Equation 4).
(3)$$I_{mixup}=\lambda I_{i}+\left(1-\lambda \right)I_{j}$$
(4)$$L_{mixup}=\lambda L_{i}+\left(1-\lambda \right)L_{j}$$
where ($I_{i},I_{j}$) are images, ($L_{i},L_{j}$) are labels of their respective images randomly selected from training data, and λ∈ [0,1]. Therefore, mixup expands the training distribution by combining prior knowledge; that is, linear interpolation of feature vectors should result in linear interpolation of related labels. Furthermore, this simple technique improves the accuracy and robustness of the proposed network while facing adversarial examples.

### 3.4. eXnet${}^{\mathrm{cm}}$(Cutout+Mixup)

In this technique, we use both operations (cutout and mixup) together during training. In order to perform more enhancement of training images, first of all, we apply cutout on training images and, after that, we use the mixup method for more generalization; a combination of these two strategies performed better by achieving the best results for all datasets.

## 4. Experiments

We have conducted extensive experiments to demonstrate the effectiveness of the proposed method in comparison with the most famous classification models including VGG19 [7], ResNet50 [20], DenseNet [47], and DeXpression [13]. All networks trained on all three datasets which are publically available and broadly used for facial emotion recognition. Detailed information about these datasets and the embedded device used in the proposed methodology is described next.

### 4.1. Datasets

#### 4.1.1. FER-2013

The most famous and widely used dataset for expression recognition was published in ICML-2013 Challenges in Representation Learning, offering more than 35,000 gray-scale images of size 48×48 with seven basic labeled emotions. For training, validation (Private_Test), and testing (Public_Test) of our methods, we utilize the division provided by ICML-2013.

#### 4.1.2. CK+

An extension of the CK dataset provides 327 labeled facial videos captured in a controlled environment. Out of the 210 facial expressions, the ethnicity of 179 is Euro-Americans and 31 belong to Afro-Americans, mostly subjects aged between 18–50 years. By extracting the last three frames from each video clip, we end up with a total of 981 images for all seven emotions. For a fair evaluation of the proposed methods, we randomly selected 10% of images from each emotion class.

#### 4.1.3. RAF-DB

Real-world Affective Faces Database (RAF-DB) is considered as one of the larger datasets having around 30,000 different facial images. This dataset provides forty annotations of each face image. The whole dataset is divided into two distinct subsets; there are seven classes in a single label subset and twelve classes in two tab subsets. Images in this collection vary greatly in ethnicity, age range, and gender. The RAF-DB dataset includes different head poses and focuses on several conditions such as lightning, occlusions, and post-processing operations. The database was divided into a training set and a test set to give an objective measure of the entries. The facial expression images in both (training and test) collections are identical. The training set consists of 80% images of the whole dataset, while 20% is used for testing.

#### 4.1.4. Embedded Device

Raspberry Pi 4B is an embedded system used to check the performance of the propose model. It gives drastic increases in processing speed, graphics, memory, and connectivity compared to the its previous generations. It can be used in robots, live applications, and smart-home applications.

### 4.2. Data Augmentation

For basic data augmentation, we apply a centre crop of size 44 × 44 and a horizontal flip for all datasets. We did not show any auxiliary task for face detection because we already resized the image by focusing only on the face. For testing, 10-crop validation, a voting mechanism, takes four crops from each corner as well as one from the center and then performs a horizontal flip on each to pass 10 crops from a single test image through the network. Finally, the classification scores are then combined to determine the most likely class.

### 4.3. Implementation Details

The implementation environment for experimentation of proposed methods consists of NVIDIA Titan RTX having 576 tensor cores, 24 GB memory, and Ubuntu 16.04 operating system. Pytorch [48] is chosen as a deep learning framework for the development of eXnet. A systematic and famous cross-entropy loss is applied. The parametric settings for the training of eXnet on each dataset are shown in Table 1. The cyclical learning rate is used only for FER-2013 and RAF-DB, starts decaying after 60 epochs, and continues decaying if loss is not improved after every five epochs. For the evaluation of CK+ dataset, we apply 10-fold cross validation. The following data augmentation techniques are applied to all datasets.

Cutout [22] follows the same details for the training as mention in Table 1, except some additional settings. Since this technique performs by cutting a portion of images from the dataset, we use the random cut of size 10×10 on images to create augmented images for increasing the number of training samples.

According to Reference [23], the author suggested to develop two different data loaders on every iteration and to mixup the images from both data loaders. However, instead of feeding two batches at the same time, we select one batch, shuffle the samples in the same batch, and then perform weighted summation. Following the same training procedure given in Reference [23], the suggested value of α is (0–0.4) for beta distribution. For emotion classification datasets, we trained the proposed network on all values of α between (0.2–1.0) but the value of α=0.6 with weight decay $1\times 10^{-4}$ revealed excellent results.

While applying cutout+mixup, initially, cutout is used on training images of all datasets as preprocessing of training samples. In order to achieve more generalized data, after cutout, mixup is also applied according to the same parametric settings mentioned above.

## 5. Results

We evaluate the proposed approach in three different ways: (i) ablation evaluation, (ii) benchmark evaluation, and (iii) real-time evaluation.

### 5.1. Ablation Evaluation

To highlight the impact of every component of eXnet on the accuracy, we trained more than 30 variants of eXnet; their results are shown in Table 2. FER-2013 dataset, with its official train, validation, and test distributions, is used to perform this study. The input shape of our model is fixed to 48×48, the same as the size of images in the predefined settings of the dataset. The following subsections define the systematic progress of the study.

#### 5.1.1. Effect of Depth Reduction on the Accuracy of the eXnet

According to deep learning, the depth of the network is directly proportional to the performance of the network. By conducting experiments on depth of network, we removed the last two convolutional layers from the network to reduce the size of the network. This reduction resulted in the form of lakes in accuracy of network, as shown in Table 2. Therefore, these layers are necessary for achieving better convergence of the network.

#### 5.1.2. Effect of Pooling Layers on the Accuracy of the eXnet

The pooling mechanism is requisite for the successful outcomes of CCNs. During this study, it is noted that, if we use convolutional layers with lower strides (1,2) in the place of pooling layers, the accuracy is decreased. However, using convolutions with higher strides (4,5) does not converge the model while the number of parameters are also increased. Therefore, it is founded that the use of pooling layers is necessary with convolutions for better outcomes of the network as mentioned in Table 2.

#### 5.1.3. Effect of Dropout on the Accuracy of the eXnet

During ablation study, dropout was not used between fully connected layers, which caused the problem of overfitting. The network reached 100% training accuracy after 20 epochs having less than 60% test accuracy. The addition of dropout between fully connected layers overcame the problem of overfitting to some extent, as displayed in Table 2.

#### 5.1.4. Effect of Kernel Size on the Accuracy of the eXnet

Table 2 demonstrates the variation of accuracy due to kernel size. From the accuracy mentioned in the table, it is clear that using higher kernel sizes cause unstable accuracy. Very high kernel size like 8, 16, and 32 do not converge with existing combinations of network parameters. The value of small kernel size (2,3) yields in the form of good convergence of network. Therefore, from the start to end of the network, the kernel size in convolutional layers is kept small.

#### 5.1.5. Effect of ParaFeat Blocks on the Accuracy of the eXnet

Table 2 shows the importance of both blocks (Parafeat) as shown in Figure 2 of the network. By using sequentially connected layers in the place of parallel feature extraction, blocks yield less accurate results. Furthermore, during depth analysis, the addition of both blocks is required for getting higher accuracy. The idea of parallel feature extraction was inspired by GoogleNet. Both Parafeat blocks create parallel features of different scales and are used for dimension reduction and refinement of extracted features. Therefore, using these blocks in our network yield impressive results.

#### 5.1.6. Effect of Fully Connected Layers on the Accuracy of the eXnet

FER-2013 is a very challenging dataset, so convolutional layers are used for feature extraction from given images. After extracting features, the input has to be classified into seven different classes. For this purposes, fully connected layers are used. On the other hand, convolutional layers are local and operated on a window of a specific size. However, for the translation of global information, fully connected layers are used. Therefore, fully connected layers help in the extraction of the global relationship between features. Table 2 shows the effect of using fully connected layers on the accuracy of the network. Without using fully connected layers, only 60% accuracy is achieved.

#### 5.1.7. Effect of Changing Optimizer on the the Accuracy of the eXnet

This subsection describes the effect of using different optimizers on the accuracy of the network. Two types of optimizers (SGD and Adam) are used, it was noticed that Adam achieved the best accuracy of 70.56% after (150) epochs. However, SGD, as compared to Adam, achieves an accuracy of 71.67% with a slightly higher (200) number of epochs, as shown in the Table 2. Hence, the SGD achieved accuracy higher than Adam.

#### 5.1.8. Effect of Learning Rate on the Accuracy of the eXnet

Table 2 demonstrates the results of both (constant and cyclical) learning rates. In the beginning, we used a constant learning rate of value 0.01, and 69% accuracy was achieved. After some experiments, it was noticed that accuracy was not improving after more than ten epochs continuously. Then, the idea of using a cyclical learning rate with a triangular strategy was used, where the learning rate fluctuates between 0.01 and 0.0001. By using this idea, we obtained a better performance of the network as compared to a constant learning rate.

#### 5.1.9. Effect of 1×1, 3×3, and 5×5 Convolutions on the Accuracy of the eXnet

At the last step, the best accuracy is achieved, with restrictions of a large size for the network. The idea of using 1×1 convolutions is used to reduce the size of the model. According to the structure of GoogleNet, 5×5 convolutional layers increase the number of parameters on a large scale; so using higher convolutions in the network design is avoided. Table 3 shows the effect of using 1×1 convolutions before 3×3 convolutional layers on the size of the network while preserving the same accuracy.

### 5.2. Benchmark Evaluation

We evaluate the proposed model on three different types of datasets for fair comparison. All three datasets include seven kinds of expressions: neutral, surprise, sad, happy, fear, disgust, and anger. We also test eXnet against existing popular classification methods; Table 4, Table 5 and Table 6 show the results evaluation for the FER-2013, CK+, and RAF-DB datasets, respectively. Our method not only outperforms other methods on all three datasets but also has less size on the disk as well. For a better comparison of the proposed approach, we trained the (VGG, ResNet, DenseNet, and DeXpression) networks under the same parametric settings as eXnet on all three datasets. The proposed shallow network performed comparatively better than these models.

According to Table 4, it can be seen that VGG19 got the highest accuracy among other benchmark networks. Still, it also has the largest size on the disk and the highest number of parameters among all models. The proposed eXnet achieves higher accuracy than VGG19, with almost 70% fewer parameters. For DeXpression and Densenet, although having fewer parameters, their accuracy is much less and the size of their model on the disk is much larger than eXnet.

During training and result collection, an important limitation of eXnet is noticed. The model has to face the problem of overfitting against all three datasets. As a solution to this problem, we apply some state-of-the-art regularization techniques [22,23]. These simple and easy to deploy techniques not only overcome the problem of overfitting but also achieved excellent accuracy on all three datasets, as shown in Figure 3, Figure 4 and Figure 5. In all three graphical representations to evaluate the performance of eXnet, part (a) indicates the performance of eXnet which clearly shows that eXnet overfit against datasets during the training process, part (b) indicates the the performance of eXnet${}^{\mathrm{cutout}}$, part (c) demonstrates the performance of eXnet${}^{\mathrm{mixup}}$, and part (d) denotes the results achieved by eXnet${}^{\mathrm{cm}}$. The visualization of our trained eXnet against face image is shown in Figure 1.

### 5.3. Real-Time Evaluation

To analyze the efficiency of the proposed model on an embedded system, the pretrained model of eXnet is deployed on Raspberry Pi 4B along with trained models of VGG, RestNet, and DenseNet. We performed experiments by inputting ten images and by calculating the mean value of time for every trained model. The results mentioned in Table 7 reveal that the proposed model also outperforms other benchmarks in a time comparison.

## 6. Conclusions

In this work, we proposed a novel CNN architecture for faster emotion classification in the wild. The ablation evaluation reveals the importance of each component for achieving higher accuracy, and the benchmark evaluation shows the superiority of our network among existing ones. Our methods not only achieves higher accuracy on the FER-2013, CK+, and RAF-DB datasets but also has less number of parameters and size on disks in comparison to the existing methods. This reduction in the number of parameters and size makes eXnet an ideal choice for embedded devices with low computational power and memory. The results of the proposed model on embedded devices also evidence that our shallow models can also be used in live applications (robots and mobile phones). In the future, we will use the “quantization” method to bring about further improvement in the efficiency of eXnet for FER in the wild.

## Figures and Tables

**Figure 1 sensors-20-01087-f001:**
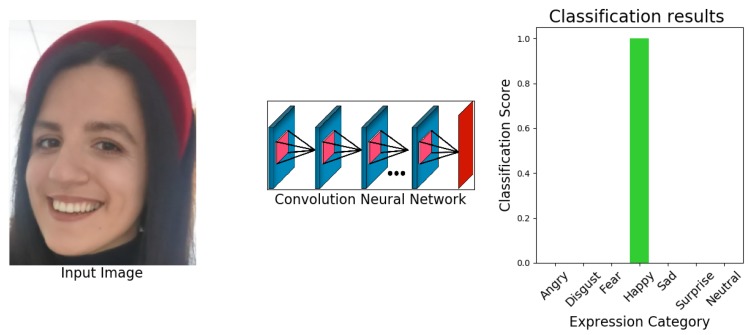
Visualization of eXnet.

**Figure 2 sensors-20-01087-f002:**
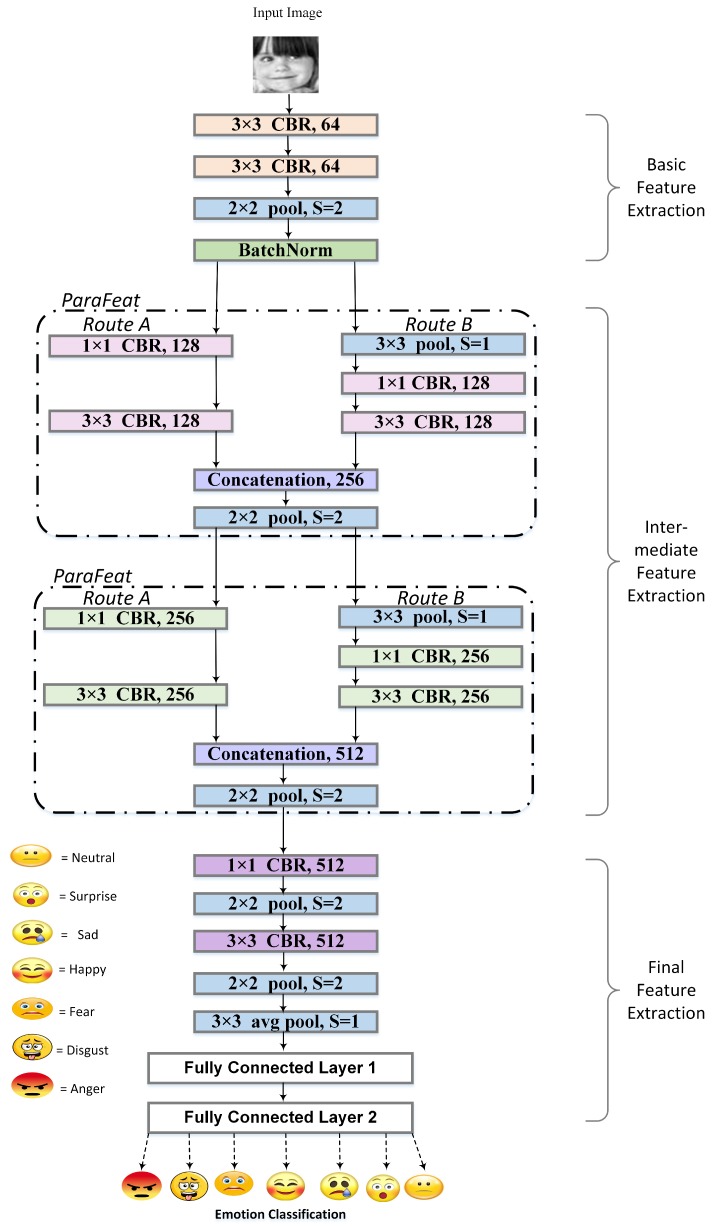
Structure of eXnet: 3×3 and 1×1 mentioned in the CBR (C = Convolutional layer, B = Batch Normalization, and R = Relu) box shows the kernel size used in convolutions, and in the pool box, S = stride.

**Figure 3 sensors-20-01087-f003:**
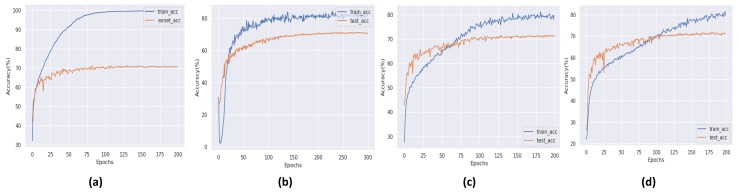
Performance of eXnet on the FER-2013 dataset.

**Figure 4 sensors-20-01087-f004:**
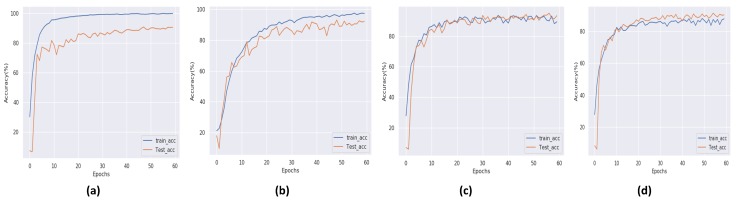
Performance of eXnet on the CK+ dataset.

**Figure 5 sensors-20-01087-f005:**
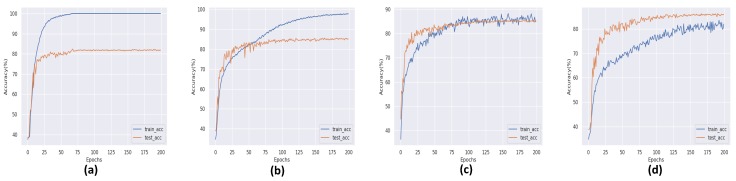
Performance of eXnet on the RAF-DB.

**Table 1 sensors-20-01087-t001:** Settings of parameters used for training of eXnet on all three dataset.

Dataset	Parameters	Values
FER-2013	Size of images used	48 × 48
	Optimizer	Stochastic Gradient Descent (SGD)
	Number of epochs	200–250
	Batch size	64
	Learning rate	0.01
	Momentum	0.9
	Learning decay	4e-5
	Cyclical learning rate	Yes
CK+	Size of images used	48 × 48
	Optimizer	SGD
	Number of epochs	60–100
	Batch size	64
	Learning rate	0.01
	Momentum	0.9
	Learning decay	4e-5
	Cyclical learning rate	No
RAF-DB	Size of images used	48 × 48
	Optimizer	SGD
	Number of epochs	150–200
	Batch size	64
	Learning rate	0.01
	Momentum	0.9
	Learning decay	1e-4
	Cyclical learning rate	Yes

**Table 2 sensors-20-01087-t002:** Ablation evaluation of eXnet.

Model Parameter	Accuracy
**Effect of depth reduction on the accuracy of the eXnet**	
eXnet after removal of initial two CBR blocks	67
eXnet after removal of last two CBR blocks	69
**Effect of pooling layers on the accuracy of the eXnet**	
eXnet having convolutions with larger strides	58
eXnet having convolutions with smaller strides	65
eXnet having pooling layers	>71
**Effect of dropout layer on the accuracy of the eXnet**	
eXnet without dropout layer	69
eXnet with dropout between fully-connected layers	71
**Effect of kernel size on the accuracy of the eXnet**	
eXnet with kernel size 32	48
eXnet with kernel size 16	57
eXnet with kernel size 8	68
**Effect of** ParaFeat **blocks on the accuracy of the eXnet**	
eXnet after skipping first ParaFeat	68
eXnet after skipping second ParaFeat	67
eXnet without using both ParaFeat	62
**Effect of fully connected layer on the accuracy of the eXnet**	
eXnet without any fully connected layer	60
eXnet without fully connected layer 1	65
eXnet without fully connected layer 2	68
**Effect of changing optimizer on the accuracy of the eXnet**	
eXnet with Adaptive moment estimation (Adam) optimizer	70
eXnet with SGD optimizer	71.67
**Effect of learning rate on the accuracy of the eXnet**	
eXnet with fix learning rate	69
eXnet cyclical learning rate	71.67

**Table 3 sensors-20-01087-t003:** Comparison between sizes of eXnet with different sizes of kernels in convolutional layers.

Model	Param
eXnet with 5 × 5 and 3 × 3 conv	20 M
eXnet with only 3 × 3 conv	12 M
eXnet with 1 × 1 and 3 × 3	4.57 M

**Table 4 sensors-20-01087-t004:** Comparison of eXnet with benchmark networks on the FER-2013 dataset.

Model	Accuracy (%)	Params	Size (MB)
VGG [7]	71.29	14.72M	70.94
ResNet [20]	71.12	11.17M	61.18
DenseNet [47]	67.54	3.0M	59.69
DeXpression [13]	68	3.54M	57.14
Liu et al. [36]	61.74	84M	-
CNN + Support Vector Machine. [9]	71	4.92M	-
Tang [11]	69.4	7.17M	-
Shao at el. [18]	71.14	7.12M	-
eXnet (ours)	71.67	4.57M	36.49
eXnet${}^{cutout}$ (ours)	71.92	4.57M	36.49
eXnet${}^{mixup}$ (ours)	72.67	4.57M	36.49
eXnet${}^{cm}$ (ours)	73.54	4.57M	36.49

**Table 5 sensors-20-01087-t005:** Comparison of eXnet with benchmark networks on the CK+ dataset.

Model	Accuracy (%) 10-Cross
VGG [7]	94.6
ResNet [20]	94
DenseNet [47]	92
DeXpression [13]	96
Tautkute et al. [10]	92
Lopes et al. [16]	92.73
Jain et al. [17]	93.24
Shao et al. [18] light-CNN	92.86 (without 10-cross )
Shao et al. [18] pretrained-CNN	95.29 (without 10-cross)
eXnet (ours)	95.63
eXnet${}^{cutout}$ (ours)	95.81
eXnet${}^{mixup}$(ours)	96.17
eXnet${}^{cm}$ (ours)	96.75

**Table 6 sensors-20-01087-t006:** Comparison of eXnet with benchmark networks on the RAF-DB dataset.

Model	Accuracy (%)
VGG [7]	82.39
ResNet [20]	81.71
DenseNet [47]	76.71
DeXpression [13]	76.33
Li et al. DLP+SVM [49]	74.20
Fan et al. [50]	76.73
eXnet (ours)	84
eXnet${}^{cutout}$ (ours)	85.59
eXnet${}^{mixup}$ (ours)	85.63
eXnet${}^{cm}$(ours)	86.37

**Table 7 sensors-20-01087-t007:** Time comparison of proposed model with benchmark networks on Raspberry Pi 4B.

Model	Time for Emotion Classification
VGG [7]	3.92 s
ResNet [20]	3.86 s
DenseNet [47]	2.09 s
eXnet (ours)	0.998 s

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
