# Peer review of "eXnet: An Efficient Approach for Emotion Recognition in the Wild"

_sensors, 2020, doi:10.3390/s20041087_

Round 1
Reviewer 1 Report
The article shows a system capable of detecting emotions from images of faces. A comparison of the proposed system is made with other systems. The proposed system is the one that gives the best results in comparison. Section 3 should be explained in more detail to clarify the proposed model.
Author Response
Author Comment: The article shows a system capable of detecting emotions from images of faces. A comparison of the proposed system is made with other systems. The proposed system is the one that gives the best results in comparison. Section 3 should be explained in more detail to clarify the proposed model.
Reviewer Response:
First of all thank you very much for your valuable comments. As reviewer mentiones that section 3 about proposed methedology can be improved. So according to this comment we tried to add more information in all three stages (3.1.1-3.1.3) to clear in more easy way. We treid to explain all the structure of proposed method in simple and easy to understand way.
Reviewer 2 Report
The study adopts a novel CNN called eXnet for facial expression recognition which outperforms other methodologies. The used datasets are FER-2013, CK+ and RAF-DB.
The paper well written and the research framework carefully presented and explained.
A minor suggestion would be to mention 3D methodologies, that are now booming.
Author Response
Reviewer Comment: A minor suggestion would be to mention 3D methodologies, that are now boomin.
Author response:
First of all i want to say a whole hearted thank you for your time and review. According to the kind suggestion of reviewer, we have added a short paragrah about emotion recognition based on 3D-CNN which is one of the famous area now a days. The last paragraph in section 2 and reference 44 both are used to fullfiled the suggestion of reviewer.
Reviewer 3 Report
The paper concern a modern approach to emotion recognition from facial expressions based on light convolutional networks.
The chosen topic is relevant and the study design and implementation correct.
The paper is generally well written. I have minor suggestions:
page 2 line 47 - "but failed to get good results" - please be more precise, good in terms of which feature? page 2, sentence lines 47-49 - please check grammar (perhaps "is" should be replaced with "being") page 2, line 68 - in order to enhance ... and stabilize page 3, line 97 - what do you mean by "shorter" model? page 7, line 209 - datasets, not dataset page 8, line 236 - what do you mean by "random crop of 40x40? page 8, line 238 - 10-crop validation - did you mean 10-fold crossvalidation? if not, specify otherwise page 8, line 238 - please describe voting mechanism used page 8, sentence in lines 249-251 is not clear, consider re-writing page 10 - descriptions in section 5.1 seem inconsistent with table 2 in two points: (1) effect of kernel size is not discussed; (2) 5.1.5 refers to what part of the table? If they are equivalent, it is not clear how. page 10, line 295 - the reference should be to table 1 or 2? page 10, line 296 - achieved page 11, line 340 - "problem raised" - it is not clear, what are you refering to.Author Response
The paper is generally well written. I have minor suggestions:
Reviewer Comment
page 2 line 47 - "but failed to get good results" - please be more precise, good in terms of which feature?
I am very thankful to this author for pointing out some typos, technical, and grammatical mistakes.
Author Response: page 2 line 47 is now corrected and clearified according to suggested opinion.
Reviewer: page 2, sentence lines 47-49 - please check grammar (perhaps "is" should be replaced with "being")
Author: page 2, sentence lines 47-49, "is" has been corrected as "being".
Reviewer: page 2, line 68 - in order to enhance ... and stabilize
Author: enhancing has been corrected to enhance and stabalizing is to stabalize on line 68, page 2.
Reviewer:page 3, line 97 - what do you mean by "shorter" model?
Author: Actually, here shorter was showing as a light-weight/shallow model. But we felt that this is not an appropriate word to show this term. So, we just changed shorter into shallower.
Reviewer: page 7, line 209 - datasets, not dataset.
Author: It was clearly a typo error, so according to suggestion line 209 is changes from dataset into datasets.
Reviewer: page 8, line 236 - what do you mean by "random crop of 40x40?
Author: From random crop mean, there is a function in python that crop an image from random position, to clarify in simple way, we cahnge line 236 to . center crop. The overall size of image was 48*48, so we crop the center part of having dimention 40*40.
Reviewer: page 8, line 238 - 10-crop validation - did you mean 10-fold crossvalidation? if not, specify otherwise page 8, line 238 - please describe voting mechanism used?
Author: This was a technical term "10-crop validation" used while testing. Therefore, we clarify this and voting mechanism used in this term in more simple and understadable way.
Reviewer: page 8, sentence in lines 249-251 is not clear, consider re-writing page 10 - descriptions in section 5.1 seem inconsistent with table 2 in two points: (1) effect of kernel size is not discussed; (2) 5.1.5 refers to what part of the table? If they are equivalent, it is not clear how.
Author: All the mistakes pointed in section 5 are carefully revised. As we also added the description about "effect of kernel" and also put added the values about "dropout". Now both of section are pretty clear.
page 10, line 295 - the reference should be to table 1 or 2
Author: This mistakes is also corrected as it was table 2.
page 10, line 296 - achieved page 11, line 340 - "problem raised" - it is not clear, what are you refering to.
Author: The Problem was overfitting, but it was not clear. According to the reviewer comment, we explain it clearly and now this problem is refering to overfitting of network over data.